# Revisiting [CLS] and Patch Token Interaction in Vision Transformers

**Alexis Marouani**[1,2][*]    **Oriane Siméoni**[1]    **Hervé Jégou**[1]    **Piotr Bojanowski**[1]    **Huy V. Vo**[1]

[1] FAIR, Meta
[2] LIGM, Ecole des Ponts, IPParis, UGE, CNRS, 77455 Marne-la-Vallée, France

## Abstract

Vision Transformers have emerged as powerful, scalable and versatile representation learners. To capture both global and local features, a learnable [CLS] class token is typically prepended to the input sequence of patch tokens. Despite their distinct nature, both token types are processed identically throughout the model. In this work, we investigate the friction between global and local feature learning under different pre-training strategies by analyzing the interactions between class and patch tokens. Our analysis reveals that standard normalization layers introduce an implicit differentiation between these token types. Building on this insight, we propose specialized processing paths that selectively disentangle the computational flow of class and patch tokens, particularly within normalization layers and early query-key-value projections. This targeted specialization leads to significantly improved patch representation quality for dense prediction tasks. Our experiments demonstrate segmentation performance gains of over 2 mIoU points on standard benchmarks, while maintaining strong classification accuracy. The proposed modifications introduce only an 8% increase in parameters, with no additional computational overhead. Through comprehensive ablations, we provide insights into which architectural components benefit most from specialization and how our approach generalizes across model scales and learning frameworks.

## 1 Introduction

In recent years, significant progress has been made in developing vision foundation models capable of generating rich and highly generalizable visual representations for images. Notably, latest state-of-the-art results have been achieved using Vision Transformer (ViT) models (Dosovitskiy et al., 2021) trained under various paradigms, including fully-supervised (Touvron et al., 2022), weakly

---

[*]Correspondence to `amarouani@meta.com`

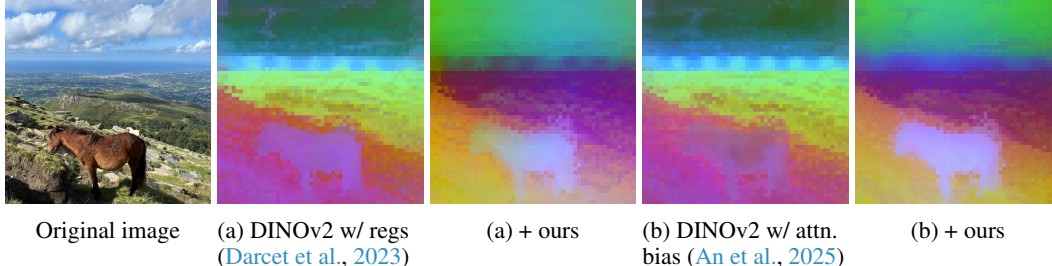

| Original image | (a) DINOv2 w/ regs (Darcet et al., 2023) | (a) + ours | (b) DINOv2 w/ attn. bias (An et al., 2025) | (b) + ours |

Figure 1: **Visualization of the impact** of our proposed layer specialization for [CLS] and patch tokens on the patch features obtained with DINOv2 when using two strategies to mitigate artifacts, namely registers ('regs') (Darcet et al., 2023) and attention bias ('attn. bias') (An et al., 2025). We display the first PCA components of model outputs in RGB.

supervised (Radford et al., 2021; Bolya et al., 2025), and self-supervised learning (Zhou et al., 2021; Oquab et al., 2023; Siméoni et al., 2025). These models capture a wide spectrum of visual semantics, enabling robust performance across a diverse range of downstream tasks and data domains.

The ViT architecture (Dosovitskiy et al., 2021) processes images by dividing them into fixed-size patches, which are then embedded and fed to a sequence of transformer blocks. Typically, a trainable class token [CLS] is prepended to the sequence of patch embeddings and is designed to aggregate information from all patches. Patches and [CLS] tokens are trained with different objectives, if any. For instance, most pre-training methods apply a loss function solely on the [CLS] token (Chen et al., 2020; Grill et al., 2020; Caron et al., 2021; Radford et al., 2021; Touvron et al., 2022). Some employs an objective on patch tokens only (He et al., 2022), while others train the [CLS] and patch tokens with separate losses (Zhou et al., 2021; Oquab et al., 2023; Siméoni et al., 2025). Regardless of the specific training paradigm, recent works (Darcet et al., 2023; An et al., 2025; Siméoni et al., 2025) show that there is a persistent imbalance between the [CLS] and patch tokens. Proposed solutions to this issue include introducing additional storage tokens into the input sequence (Darcet et al., 2023), modifying the attention mechanism (An et al., 2025), or incorporating additional loss terms to explicitly constrain patch locality (Siméoni et al., 2025). In contrast, we hypothesize that the imbalance arises because models process the [CLS] and patch tokens through identical computational pipelines, despite their fundamentally different roles and nature, and propose disentangling their treatment to overcome the imbalance.

In this work, we analyze the model statistics in order to better understand the internal mechanisms that govern the interaction between the [CLS] and patch tokens. Our analysis reveals a surprising finding: normalization layers are already implicitly learning to distinguish between the [CLS] and patch tokens before the attention mechanism. Building on this insight, we introduce a simple yet effective architectural modification that explicitly separates the processing of the [CLS] and patch tokens, as illustrated in Fig. 5. With just a minimal set of specialized layers, our approach leads to noticeably richer dense features (see Fig. 1) and delivers substantial gains on dense prediction tasks. For instance, we improve the average mIoU scores on segmentation benchmarks by as much as 2.2 points with a ViT-L. This work sheds light on hidden dynamics within transformer models and also demonstrates how targeted architectural changes can translate into significant real-world performance improvements. We make the following contributions:

- We analyze the interactions between [CLS] and patch tokens within Vision Transformers, and show that models implicitly attempt to distinguish them through normalization layers.
- We propose an architectural modification that specializes their computations to reduce the friction between them, while keeping the number of operations constant. We study different specialization strategies for transformer block components.
- We demonstrate the generalizability of our approach across model scales and learning frameworks, showing significant improvements in dense prediction tasks without compromising classification performance.

## 2 RELATED WORK

**Vision Transformers** Inspired by Vaswani et al. (2017) and first introduced by Dosovitskiy et al. (2021), Vision Transformer has become an architecture of choice when building vision models. A typical ViT model consists of a patch embedder and a stack of transformer blocks. Given an image, the patch embedder divides it into equally-sized square patches and transforms them into patch tokens that represent local information in the image. Optionally, a learnable [CLS] token is added to the set of patch tokens in order to capture global information. All tokens are then passed through the transformer blocks which process them with various transformations, most notably the multi-head self-attention operator (Vaswani et al., 2017) that allows tokens to attend to each other. Built on the original architecture, subsequent works have introduced additional components to improve various aspects of ViTs such as data efficiency (Touvron et al., 2020; Yuan et al., 2021), computational cost (Liu et al., 2021; Bolya et al., 2023) and normalization (Touvron et al., 2021). ViT architecture has enabled state-of-the-art performance in various tasks (Carion et al., 2020; Strudel et al., 2021), simplified multi-modal learning (Radford et al., 2021; Fini et al., 2024), and led to excellent local and global representation in foundation models (Oquab et al., 2014; Tschannen et al., 2025; Siméoni et al., 2025). In most ViTs, [CLS] and patch tokens are functionally interchangeable in transformer

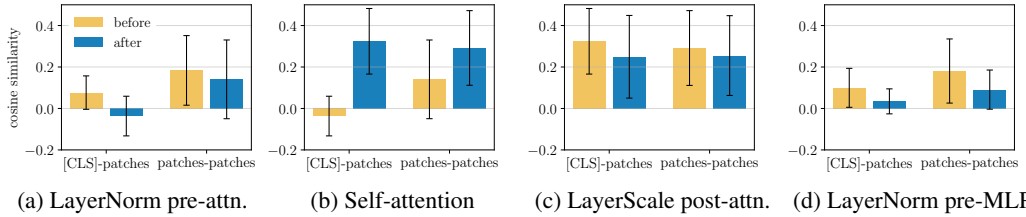

(a) LayerNorm pre-attn.  (b) Self-attention  (c) LayerScale post-attn.  (d) LayerNorm pre-MLP

Figure 2: **`[CLS]`-patches separation effect within transformer blocks** in vanilla DINOv2 ViT-L model. We show mean and standard deviation of cosine similarity between `[CLS]` and all patches, and all-to-all patches, before and after each transformer layers. 'attn.' stands for attention.

blocks – they are processed in identical manner using the same operators – despite their distinctive nature. We show in our analysis that the identical treatment of these tokens is suboptimal and that disentangling them leads to better local features for dense tasks.

**Improving dense feature learning**  Visual representation learning approaches have mostly focus on optimizing the global representation by primarily training the `[CLS]` token to summarize the image content either in supervised (Touvron et al., 2022), weakly supervised (Radford et al., 2021; Tschannen et al., 2025) or self-supervised settings (Caron et al., 2021; Liu et al., 2021). As a by-product, they also produce local representation that perform well on tasks that require fine-grained features such as object detection, semantic segmentation or depth estimation. Most notably, the self-supervised method DINO (Caron et al., 2021) produces excellent patch features that supercharge research on unsupervised object detection and segmentation. iBoT (Zhou et al., 2021) augments DINO with masked image modeling (He et al., 2022) to optimize both global and local representation. DINOv2 (Oquab et al., 2023) introduces new technical components such as Sinkhorn-Knopp centering and untying heads to successfully scale DINO to large datasets and model sizes, achieving excellent performance on dense tasks. Learning meaningful dense features with Vision Transformers is not without challenges. Darcet et al. (2023) discusses the noisy attention maps produced by models trained at scale during longer training periods. This issue, which degrades dense prediction performance, is caused by some patch tokens losing their local context after being repurposed by the model to store global information. They propose an architectural solution with registers to mitigate these issues. Other successful attempts to enhance the quality of local features include regularizing similarity to neighbor patches post-training (Pariza et al., 2024) or recovering patch similarity with Gram anchoring mechanism (Siméoni et al., 2025). Similar to these works, we improve dense features quality during training by specializing `[CLS]` and patch tokens treatment within the Transformer blocks of ViTs and thus reducing the friction between them.

## 3    FRICTION BETWEEN [CLS] AND PATCHES

Vision Transformers are typically trained with a trainable `[CLS]` token which encodes global information about the image prepended to the sequence of patch tokens. Despite the distinct nature of the `[CLS]` and patch tokens, current models treat them equivalently, applying the exact same operations to both. However, (Darcet et al., 2023) has highlighted potential communication issues between these two types of tokens, leading to a severe loss of locality of patch tokens and the appearance of undesirable outliers in the attention maps. While registers help to mitigate the appearance of artifacts, we argue that more could be done. Our observations indicate that a degree of friction persists between `[CLS]` and patch tokens, as discussed below.

**ViTs differentiate `[CLS]` and patch tokens for the attention**  We analyze the interplay between `[CLS]` and patch tokens by computing their similarity at different points within the model, before and after principal layers in each transformer block. In Fig. 2, we visualize the mean and standard deviation of these similarities. Our results are averaged over patches of 1000 images and across all model blocks. Additionally, we present the same statistics between patches. While certain operations—such as the LayerScale applied post-attention—have little effect on the similarity between `[CLS]` and patch tokens, the self-attention layer markedly increases their similarity. This increase is expected, as self-attention realigns the different token types. However, our analysis uncovers a surprising phenomenon: the representations of `[CLS]` and patch tokens naturally diverge at specific

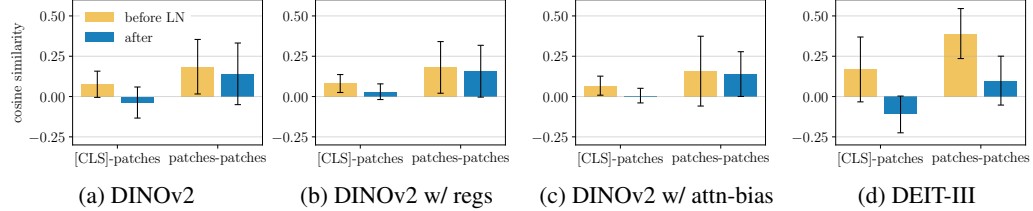

| (a) DINOv2 | (b) DINOv2 w/ regs | (c) DINOv2 w/ attn-bias | (d) DEIT-III |

Figure 3: **Impact of LayerNorm before attention layer** for different pre-trained models. We show mean and standard deviation of cosine similarity between [CLS] and all patches, and between all patches. Statistics visualized before and after LayerNorm (LN).

stages of the computational pipeline, particularly just before attention operations. Indeed, the LayerNorm applied before attention drastically reduces the similarity between [CLS] and patch tokens, bringing it close to zero. This implicit differentiation indicates that the model attempts to adapt to the distinct functional roles of these token types before the attention mechanism, despite their shared parameterization. We plot the statistics of more layers in Sec. A.2.

**Role of the pre-attention LayerNorm** In Fig. 3, we focus on the impact of the pre-attention LayerNorm to the similarity between the [CLS] and patch tokens in different pre-trained models, including DINOv2 (Oquab et al., 2023) and its variants with registers (Darcet et al., 2023), noted 'regs', and attention bias (An et al., 2025), noted 'attn. bias', and supervised DEIT-III (Touvron et al., 2022). It can be observed that in all cases, prior to the attention mechanism, the LayerNorm disentangles the [CLS] and patch tokens, enabling them to serve distinct functions within the attention process. This phenomenon appears in all pre-trained models with different extent. For instance, the LayerNorm strongly enforces negative correlation between [CLS] and patches in DINOv2 and DEIT-III while keeping the correlation close to zeros in the variants of DINOv2. In contrast, the similarity among patch tokens remains positive and largely stable, with only a slight decrease observed—a phenomenon we interpret as a regularization effect. This effect likely prevents rank collapse and promotes a more uniform distribution of tokens on the unit sphere, consistent with observations reported in Wu et al. (2024).

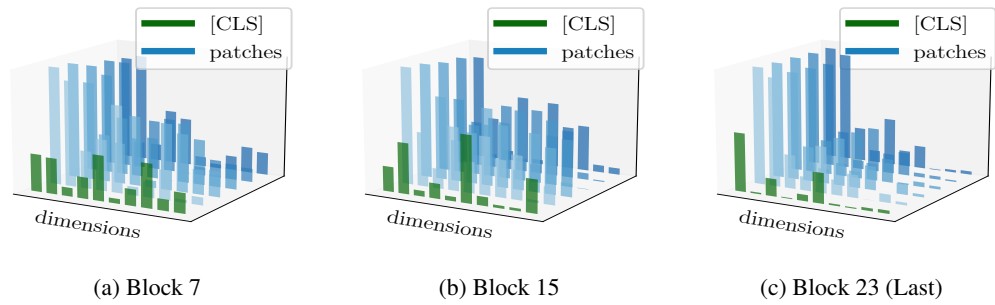

| (a) Block 7 | (b) Block 15 | (c) Block 23 (Last) |

Figure 4: **Dimensions with biggest magnitudes** early (a), in the middle (b), at the end (c) of the model for [CLS] and 5 patches with the highest magnitudes in the selected dimensions. Tokens taken at the output of blocks. The considered model is a DINOv2 ViT-L with attention bias.

**Dimension separation** To understand how a separation effect can appear in a LayerNorm layer, one has to recall that it performs a point-wise normalization and a dimension-wise affine transformation. Therefore, a separation effect occurs when inputs have very different magnitudes in each dimension. In Fig. 4, we plot the dimensions with biggest absolute magnitudes—averaged over patches and [CLS]—at the output of different blocks. We observe that some specific dimensions are leveraged only by a certain token type. For example, in Fig. 4c, the 2nd dimension presents large magnitudes for patches and almost none for [CLS]. Moreover, the deeper we are in the model, the fewer token types share dimensions. This enables normalization layers to perform distinctive operations. More than just regularizing, they specialize and separate the tokens.

All the observations above indicate that treating [CLS] and patch tokens identically compels the model to allocate resources towards implicitly separating them, which could be used to learn more

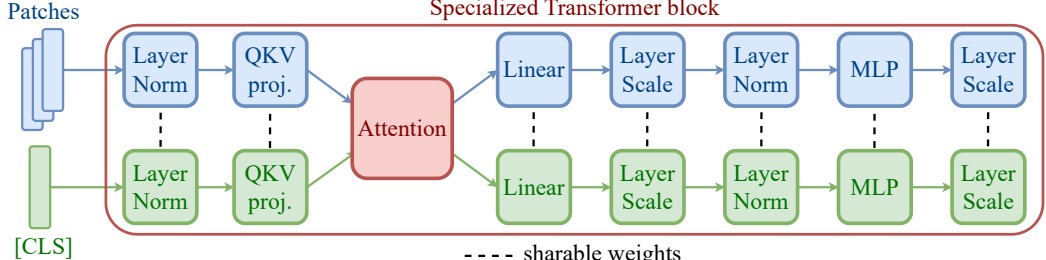

Figure 5: **Architecture specialization.** We investigate how [CLS] and patch tokens can be processed through specialized layers, while preserving their interactions within the attention mechanism.

meaningful features. We argue that disentangling their treatments would facilitate the model in learning better representation, as discussed in next section.

## 4   [CLS] - PATCHES SPECIALIZATION: ANALYSIS

In this section, we first define our proposed layer specialization in Sec. 4.1 and set the experimental setting in Sec. 4.2. In Sec. 4.3, we discuss the benefits of splitting normalizations for [CLS] and patch tokens. We also investigate which part of the model needs specialization in Sec. 4.4, and more specifically which layers in Sec. 4.5.

### 4.1   OUR PROPOSAL: LAYER SPECIALIZATION

Based on observations made in the previous section, we explore disentangling the computation of global and local representations in ViTs. Taking inspiration from the success of double-stream architectures to handle different modalities (Esser et al., 2024), we explore a similar approach for the [CLS] and patch tokens. More specifically, inside a classic transformer block, [CLS] and patch tokens go through several layers: projections, some normalizations and a MLP. We propose to decouple the [CLS] and patch tokens by processing them with different weights for certain layers. Indeed, instead of using a single layer to process both token types, we introduce two distinct layers—each with its own set of weights—specialized for either [CLS] or patch tokens. This allows each layer to better capture the unique characteristics of its respective token type. However, the tokens continue to interact through the attention mechanism as usual, ensuring information flow between [CLS] and patch tokens is preserved. An illustration of this specialized architecture is provided in Fig. 5. While this approach introduces some additional memory overhead, our experiments show that the increase in model size remains small—approximately 8%—when layer specialization is applied selectively to achieve optimal performance. More importantly, layer specialization does not increase inference FLOPs, as the model continues to perform the same computational operations during inference. This ensures that the efficiency of the model is maintained, even as we enhance its representational capacity through targeted specialization.

### 4.2   EXPERIMENTAL SETTING: TRAINING AND EVALUATION

**Training**   We investigate layer specialization with different pre-training paradigms including the popular self-supervised strategy DINOv2 (Oquab et al., 2023) and the fully-supervised DeiT-III (Touvron et al., 2022). We also investigate different model sizes (ViT-B, L, H). Unless specified otherwise, we produce results with a ViT-L model trained following DINOv2 recipe. Following An et al. (2025), we integrate the attention bias strategy, which mitigates high-norm anomalies (Darcet et al., 2023) without introducing additional tokens, in all models and attention operations. More discussion can be found in Sec. A.1. For DINOv2, we train our models on ImageNet-22K (Ridnik et al., 2021) dataset for 600k training steps. For DeiT-III, we train our models on ImageNet-1K (Deng et al., 2009) for respectively 400 epochs on ViT-B and 800 epochs on ViT-L. For both training paradigms, we pre-train using the first pre-training phase, and drop the high-resolution fine-tuning step. We report more details in Sec. A.3.

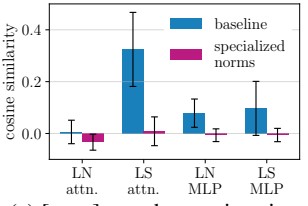
(a) [CLS] -patches cosine sim.

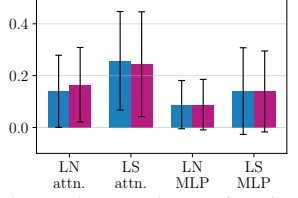
(b) Patches-patches cosine sim.

| Spec. | Linear Acc. | Avg Seg. | Avg Depth ↓ |
|---|---|---|---|
| – | **85.4** | 64.5 | 1.232 |
| norms | 85.1 | **65.6** | **1.178** |

(c) Norm specialization

Figure 6: **Normalization specialization.** Mean and standard deviation of the cosine similarity computed between (a) [CLS] and all patches and (b) all to all patches. We compare post-normalization statistics between the baseline architecture (DINOv2 ViT-L with attn. bias) and when specializing the normalization layers for [CLS] and patch token (specialized norms). (c) Quantitative results with specialized normalization layers. 'LN' stands for LayerNorm and 'LS' for LayerScale.

**Evaluation**  Following Oquab et al. (2014), we assess model representations via linear probing on global, with ImageNet-1k (Deng et al., 2009), and dense prediction tasks. For semantic segmentation, we use ADE20K (Zhou et al., 2017), Cityscapes (Cordts et al., 2016) and PASCAL VOC (Everingham et al., 2010), reporting mIoU. For depth estimation, we use KITTI (Geiger et al., 2013), NYU Depth v2 (Nathan Silberman & Fergus, 2012) and SUN RGB-D (Song et al., 2015), reporting RMSE. For detection, we use COCO (Lin et al., 2014), reporting AP. Some tables show average segmentation and depth scores across corresponding benchmarks. More details in Sec. A.3.

## 4.3 SPECIALIZING NORMALIZATION LAYERS

As discussed in Sec. 3, ViTs attempt to separate the [CLS] and patch tokens with the LayerNorm applied prior to the attention operation. Building on this observation, our initial experiment focuses on specializing the normalization layers (LayerNorms and Layer Scales) within the model, with the aim of further supporting the model's inherent tendency to separate these feature types.

We specialize the normalization layers in all blocks of the model. This lightweight modification introduces only 0.05% additional parameters, yet significantly alters the feature distributions. In Fig. 6a, we report the mean and standard deviation of the cosine similarity between the [CLS] and patch tokens, computed after each normalization layer. We compare our variant with specialized normalization weights to the baseline. Conversely, Fig. 6b shows the corresponding statistics when using all patches instead of the [CLS] . We observe that specializing the normalization layers further amplifies the disentanglement of the [CLS] and patch tokens, resulting in a more distinct separation of their embeddings after each normalization step.

The impact of these specialized normalizations is quantified in Fig. 6c. The specialization leads to significant improvements on dense prediction tasks, yielding an average increase of $+1.1$ mIoU points on segmentation benchmarks and an improvement of $-0.054$m on depth estimation. These results highlight that a better specialization of the token types benefits the patches representations. On the other side, global results are slightly degrading. We however show in the next section that this loss can be mitigated. Unless otherwise specified, in the remainder of the paper, we apply specialized normalization layers to all transformer blocks.

## 4.4 BLOCK-LEVEL TARGETED SPECIALIZATION

While normalization layers in ViTs show in overall a [CLS] -patch separation effect, we have observed that the extent of their impact is not uniform across all blocks. It can be seen from Fig. 7, which depicts [CLS] -patches cosine similarity before and after the first LayerNorm in each block, that the separation effect of the normalization varies depending on its position within the model. Indeed, blocks at the beginning and near the end see the most impact. We hypothesize that the importance of separation in the early blocks stems from their proximity to the different inputs. Although the [CLS] token is trained to summarize information from the patches, it is initialized as a learned parameter and thus has an input distribution very distinct from that of the patch tokens. Later in the model, separation becomes important again as tokens are closer to the final representations and the training objectives. The observations above suggest that we can benefit from more targeted

| Specialized blocks | Linear Acc. | Avg Seg. | Avg Depth ↓ |
|---|---|---|---|
| ∅ | **85.4** | 64.5 | 1.232 |
| 1st half | 85.1 | 65.7 | **1.191** |
| 2nd half | 85.2 | 64.5 | 1.236 |
| All | 85.0 | **65.9** | **1.191** |

(a) Which model's part to specialize

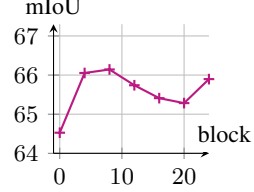

(b) Avg. segmentation perf.

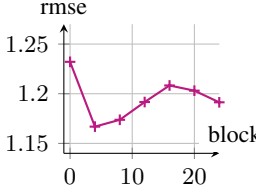

(c) Avg. depth perf. ↓

Figure 8: **Block specialization.** (a) Performance metrics when specializing specific parts of the model (b) Average segmentation scores and (c) average depth rmse (↓) vs number of specialized blocks at the beginning of the model. Normalization layers are specialized in all blocks. The base model is a DINOv2 ViT-L with attention bias.

specialization within the model. We study next which blocks should be specialized to optimize the model's performance.

We first quantitatively compare the impact of specializing different sections of the model, as shown in Fig. 8a. To this end, we train DINOv2 while specializing either the first half, the second or all of the transformer blocks, on top of specializing all normalization layers. Within a block, all layers are specialized. Our findings indicate that the best performance is achieved when specializing the early layers, which are closest to the input. Specifically, specializing the first half of the layers improves the segmentation results by an average of 1.2 mIoU points, with only a negligible decrease in linear accuracy. In contrast, specializing the late layers yields no improvement compared to the baseline. We attribute this to the fact that [CLS] and patch tokens share the representation space in the first part; once this interaction is established, further special-

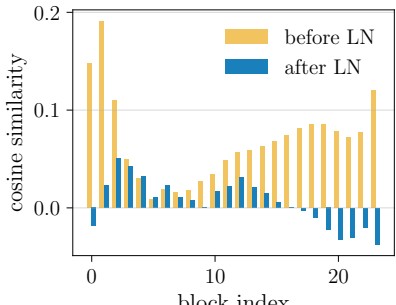

Figure 7: Mean cosine similarity between [CLS] and all patches before and after the first LayerNorm (LN) of each block.

ization has limited effect. Finally, while specializing all layers produces the highest segmentation performance, it comes with a higher memory cost and a larger drop in linear accuracy.

We further analyze how the number of specialized blocks, starting from the first, affects performance (Fig. 8b and 8c). We vary the number of blocks specialized from 0 to 24 (total number of blocks in ViT-L) in steps of 4, and observe that specializing the first third of the model yields the best results, while specializing later layers degrades performance. Notably, the optimal point at one third of the model's depth coincides with a marked shift in the statistics of similarity scores shown in Fig. 7, which might explain the effectiveness of specializing the early layers.

## 4.5 TARGETED SPECIALIZATION WITHIN TRANSFORMERS BLOCKS

The previous section has shown that careful selection of transformer blocks for specialization is important for optimizing the performance. We now explore whether further improvements can be achieved with a targeted selection of *specific layers* to specialize within the transformer blocks. In the following experiments, we specialize different layers of blocks in the first third of the model while also applying specialization to the normalization layers in all blocks.

Table 1 shows model performance on global and dense prediction tasks when specializing different layers (QKV projection, Linear and MLP, see Fig. 5). We observe that the performance on global task remain largely stable independently of the selected layers. In contrast, results on dense segmentation tasks get further improvements beyond what is achieved with normalization specialization alone. Interestingly, the gains do not increase monotonically with the number of specialized layers or additional parameters, as might be expected from typical scaling laws (Touvron et al., 2021). Specializing either or both QKV and post-attention projections consistently yields improvements. In particular, the greatest performance gains are achieved by specializing the QKV projection, which

Table 1: **Layer specialization ablation.** Performance and increase in parameter count of models trained with different layer specialization strategies, applied to the first third of the transformer blocks. In all cases, the normalization specialization described in Sec. 4.3 is applied in all blocks. The base model is a DINOv2 ViT-L with attention bias.

| QKV proj. | Linear | MLP | Parameter Increase (%) | Linear Accuracy | Avg. Seg. | Avg. Depth ↓ |
|---|---|---|---|---|---|---|
| | | | 0.05 | 85.1 | 65.6 | 1.178 |
| | ✓ | | 3 | **85.3** | 66.1 | 1.191 |
| ✓ | | | 8 | 85.2 | **66.6** | 1.165 |
| ✓ | ✓ | | 11 | 85.1 | 66.1 | 1.180 |
| | | ✓ | 22 | 85.1 | 65.2 | 1.189 |
| | ✓ | ✓ | 25 | 85.1 | 65.9 | **1.163** |
| ✓ | | ✓ | 30 | **85.3** | 65.6 | 1.185 |
| ✓ | ✓ | ✓ | 33 | **85.3** | 66.1 | 1.174 |

Figure 9: **Performances and training dynamics.** Performance vs. training iterations on global task ImageNet classification (IN1k) and dense tasks—segmentation (ADE20k, Cityscapes, VOC) and depth (KITTI, NYU)—with linear probing. We compare baseline DINOv2 ViT-L with attn. bias and when specializing QKV projection in the first $1/3$ of the model and all normalizations.

introduces only $8\%$ additional parameters while delivering an average increase of $+1$ mIoU point over normalizations alone. In contrast, specializing the post-attention projection does not offer further benefits, and specialization of the MLP layer either has no effect or negatively impacts performance. Note that we could ease this $8\%$ memory cost overhead with Low Rank Adaptation used when specializing the QKV projection. We produce encouraging preliminary results with different ranks in Sec. A.5 and leave further investigation as future work.

Our overall results show that increasing the disentanglement between [CLS] and patch tokens before the attention mechanism (with separated normalizations and projection) contributes to improved dense prediction performance. We hypothesize that encouraging the [CLS] and patch tokens to assume more distinct roles in the attention mechanism enhances their interactions, ultimately improving overall model effectiveness. We also report results in Sec. A.6 for the setting where only the QKV projections are specialized, while the normalization layers remain shared. In this configuration, performance is comparable to the baseline, indicating that the specialization of normalization layers is critical to achieve improvements, as shown in Sec. 4.3.

In Fig. 9, we compare performance dynamics of normalization and QKV projection specialization against baseline DINOv2 with attention bias. Across all dense benchmarks—of both segmentation and depth estimation—specialization consistently enhances results. These improvements are evident from early stages of training and continue to increase over time. This trend suggests that employing specialization not only boosts performance but also contributes to more stable training dynamics.

Table 2: **Generalizability** of the specialization on (a) DINOv2 when using different high-norm handling strategies (4 registers (Darcet et al., 2023), attention bias (An et al., 2025) ('attn. bias') or none (∅)), (b) different ViT sizes on DINOv2 with attention bias framework and (c) a supervised framework: DeiT-III. Relative difference between baseline and our specialization ('+ours') is shown in green if improvement and red otherwise.

| Method | Size | Classif. | Segmentation | | | Depth ↓ | | | Detection |
|--------|------|----------|--------------|---|---|---------|---|---|-----------|
| | | ImNet | ADE | City | VOC | KITTI | NYU | SUN | COCO |
| *DINOv2 - With high-norm handling strategies* | | | | | | | | | |
| ∅ | L | 85.3 | 45.7 | 64.2 | 82.1 | 2.868 | 0.389 | 0.410 | 45.6 |
| +ours | | 85.3+0.0% | 47.3+3.5% | 66.6+3.7% | 83.7+1.9% | 2.787-2.8% | 0.369-5.1% | 0.390-4.9% | 46.8+2.6% |
| 4 registers | L | 85.3 | 45.6 | 64.9 | 82.2 | 2.893 | 0.372 | 0.411 | 46.4 |
| +ours | | 85.3+0.0% | 47.5+4.2% | 65.9+1.5% | 83.6+1.7% | 2.906+0.4% | 0.367-1.3% | 0.395-3.9% | 46.8+0.9% |
| Attn. bias | L | 85.4 | 46.2 | 65.2 | 82.2 | 2.917 | 0.373 | 0.406 | 46.0 |
| +ours | | 85.2-0.2% | 48.4+4.8% | 67.4+3.4% | 84.0+2.2% | 2.739-6.1% | 0.362-2.9% | 0.393-3.2% | 48.2+4.8% |
| *DINOv2 - Other sizes* | | | | | | | | | |
| Attn. bias | B | 80.4 | 38.3 | 58.4 | 76.6 | 3.250 | 0.462 | 0.464 | 39.6 |
| +ours | | 80.6+0.2% | 38.5+0.5% | 60.3+3.3% | 76.5-0.1% | 3.236-0.4% | 0.448-3.0% | 0.470+1.3% | 39.8+0.8% |
| Attn. bias | H | 86.2 | 48.1 | 67.0 | 83.1 | 2.717 | 0.359 | 0.387 | 49.9 |
| +ours | | 86.1-0.1% | 49.2+2.3% | 67.1+0.1% | 83.5+0.5% | 2.752+1.3% | 0.344-4.2% | 0.386-0.3% | 49.5-0.8% |
| *DeiT-III* | | | | | | | | | |
| Attn. bias | B | 81.8 | 25.4 | 61.7 | 48.9 | 5.040 | 0.747 | 0.823 | 32.8 |
| +ours | | 81.7-0.1% | 26.3+3.5% | 62.7+1.6% | 50.7+3.7% | 4.900-2.8% | 0.732-2.0% | 0.809-1.7% | 32.5-0.9% |

## 4.6 GENERALIZATION RESULTS

We investigate the generalizability of our specialization approach across different variants of DINOv2, as presented in the upper part of Table 2. Specifically, we train models using the DINOv2 recipe with two high-norm handler strategies—registers (Darcet et al., 2023) ("4 registers") and attention bias (An et al., 2025) ("attn. bias")—as well as without any handler. In all cases, we observe that specialization consistently boosts dense prediction results by up to 4.8% on ADE20k, while having a negligible effect on classification performance (no decrease greater than 0.2%). This shows that better separating the treatment of `[CLS]` and patch tokens is complementary to both high-norm handling strategies to improve dense features. We also investigate the specialization on DINOv2 ViT-B and ViT-H models and present results in the middle section of Table 2. It can be seen that our proposed specialization leads to improvements on most benchmarks, confirming its generalizability across different ViT model sizes.

We further explore the fully-supervised training setting by applying specialization to a ViT-B trained with DEIT-III (Touvron et al., 2022) strategy. We observe consistent improvements in dense prediction tasks, with gains reaching up to 3.7% on VOC. For ViT-L, specialization does not yield benefits, likely due to the training dynamics related to the absence of a local loss to guide dense feature learning which causes the dense performance to degrade over time (we provide more details in Sec. A.7). These results suggest that the effectiveness of the specialization may depend on training objectives, highlighting promising directions for future research.

Finally, we visualize the learned patch representations using PCA in Fig. 1 for DINOv2 models trained with either registers or attention bias. In both settings, incorporating our specialization strategy produces cleaner and more semantically meaningful patch representations. Specifically, this approach reduces artifacts in textures and uniform regions, resulting in more accurate object segmentation. More visualizations can be found in A.8.

## 4.7 MORE COMPARISON

We compare our specialization approach to the class-attention mechanism from CaiT (Touvron et al., 2021) in Table 3. We train both models following the DINOv2 framework, with and without attention bias. In CaiT architecture, patch tokens are processed through the transformer blocks, then the `[CLS]` is appended and updated via 2 class-attention layers to aggregate information from the patch

Table 3: **CaiT vs specialization** results. ViT-L models trained with DINOv2 framework. We compare DINOv2 baseline (∅) with class-attention (CaiT) and specialization (ours) architectures. Results without and with attention bias (Attn. bias). Best result in bold.

| Method | Classif. | Segmentation | | | Depth ↓ | | | Detection |
|---|---|---|---|---|---|---|---|---|
| | ImNet | ADE | City | VOC | KITTI | NYU | SUN | COCO |
| ∅ | **85.3** | 45.7 | 64.2 | 82.1 | 2.868 | 0.389 | 0.410 | 45.6 |
| CaiT | 84.0 | 43.5 | 63.7 | 80.8 | 2.968 | 0.397 | 0.421 | 43.1 |
| Ours | **85.3** | **47.3** | **66.6** | **83.7** | **2.787** | **0.369** | **0.390** | **46.8** |
| Attn. bias | **85.4** | 46.2 | 65.2 | 82.2 | 2.917 | 0.373 | 0.406 | 46.0 |
| Attn. bias + CaiT | 85.2 | 45.3 | 65.7 | 82.0 | 2.882 | 0.374 | 0.400 | 46.8 |
| Attn. bias + ours | 85.2 | **48.4** | **67.4** | **84.0** | **2.739** | **0.362** | **0.393** | **48.2** |

tokens of the last block. We report results in Table 3, we observe that our specialization consistently outperforms the class-attention mechanism across all downstream tasks.

## 5 CONCLUSION

In this work, we investigate the disentanglement of [CLS] and patches computations in Vision Transformers, focusing on their distinct roles and interactions. Through a comprehensive analysis, we demonstrate that disentangling their processing pathways and selectively specializing architectural layers leads to significant improvements in dense prediction tasks, including segmentation and depth estimation, while maintaining strong global performance. Our approach achieves these gains without increasing computational overhead, with minimal additional parameter cost, and generalizes across multiple ViT architectures and frameworks. These findings highlight the importance of tailored architectural designs and suggest promising directions for future research, including further exploration of efficient specialization strategies and applications to broader modalities and tasks.

ACKNOWLEDGMENTS

We thank Francisco Massa and Maximilian Seitzer for their insightful discussions and help during the course of this work.

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

# A   APPENDIX

## A.1   ADDRESSING TOKEN INTERACTION ANOMALIES

In this work, we examine how the distinct roles of [CLS] and patch tokens affect their interactions within the model. Previous works (Darcet et al., 2023; Sun et al., 2024) show that despite sharing computational pathways, these different token types develop inter-dependencies that can lead to token anomalies, manifested as high-norm outliers in the patch features space. These anomalies suggest an underlying tension in how information flows between global and local representations.

**Registers.**   In order to mitigate such artifacts, observed when using different pre-training strategies (Oquab et al., 2023; Touvron et al., 2022; Radford et al., 2021), Darcet et al. (2023) propose to add learnable register tokens to the input sequence, whose roles are to replace the high-norm patches in the internal communication between patches and the [CLS] token. Doing so mitigates the appearance of such artifacts and boost overall results.

**Attention bias.**   The recent study on artifacts in Large Language Models by An et al. (2025) investigates the systematic appearance of outliers which they link to the attention mechanism. They propose a solution consisting in adding learnable biases to the keys and values in each attention head. They analyze the equivalence of their solution compared to registers.

Table 4: **The impact of norm handling strategies** on DINOv2 results.

| Norm. method | IN | ADE | City. | NYU↓ |
|---|---|---|---|---|
| ∅ | 85.3 | 45.7 | 64.2 | 0.389 |
| 4 registers | 85.3 | 45.6 | 64.9 | **0.372** |
| attn. bias | **85.4** | **46.2** | **65.2** | 0.373 |

In our experiments, we observe that both strategies have a similar impact on high-norm artifacts and as seen in Table 4, best overall performance is achieved when using the attention bias ('attn. bias') strategy, with a significant improvement on segmentation benchmarks (e.g. ADE20k and Cityscapes). To minimize confounding factors that could affect the interaction between the [CLS] and patch tokens, we adopt the attention bias strategy, which mitigates high-norm anomalies without introducing additional tokens.

## A.2   EFFECT OF OTHER LAYERS ON [CLS] -PATCHES SIMILARITIES

We report in Fig. 10, the effect on [CLS] -patches similarity of the MLP and post-MLP LayerScale layers within transformer blocks. The MLP layer, similar to the self-attention layer, increases the similarity between [CLS] and patches as it aligns the features. The post-MLP LayerScale, similar to other normalization layers, shows a stronger disentangling effect.

## A.3   TRAINING AND EVALUATION DETAILS

Throughout this work, we follow the experimental protocol of Oquab et al. (2023) and evaluate the performance of the trained models on a set of global and dense task benchmarks.

**Classification**   For the global task, we perform linear probing on ImageNet classification (Deng et al., 2009). We train a linear layer with SGD for 12500 iterations, using random-resized-crop data augmentation, and the [CLS] token as input for the linear layer. We also perform the following grid search on learning rate : $\{1.0e^{-5}, 2.0e^{-5}, 5.0e^{-5}, 0.0001, 0.0002, 0.0005, 0.001, 0.002, 0.005, 0.01, 0.02, 0.05, 0.1\}$

We then report the highest accuracy value obtained on the validation set as is common practice.

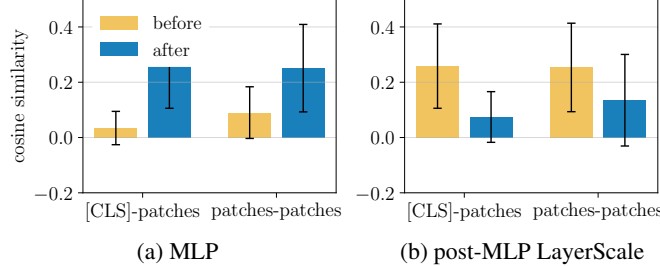

Figure 10: **Effect of the MLP and post-MLP LayerScale layers on `[CLS]`-patches similarity** in vanilla DINOv2 pre-trained model. We show mean and standard deviation of the cosine similarity between `[CLS]` and all patches ('CLS-patches'), and between patches ('patches-patches'), before and after the considered layers.

**Segmentation** For semantic segmentation, we use ADE20K (Zhou et al., 2017), Cityscapes (Cordts et al., 2016), and VOC (Everingham et al., 2010), and report the mean Intersection over Union (mIoU) scores for each. When we report the average performance of segmentation tasks, we average the scores across these 3 datasets. We train a linear classifier on the training set of each benchmark for 40000 iterations with a learning rate of $1e^{-3}$. This linear layer is applied on top of the patch output features (after the last layer normalization) of the frozen backbone, with the features further normalized using a trained batch normalization layer.

**Depth estimation** For depth estimation, we evaluate on KITTI (Geiger et al., 2013), NYU Depth v2 (Nathan Silberman & Fergus, 2012), and SUN RGB-D (Song et al., 2015), reporting the average Root Mean Squared Error (rmse) scores. When we report the average performance of dense tasks, we average the scores across these 3 datasets. We train a linear classifier on the training set of each benchmark for 38400 iterations with a learning rate of $1e^{-3}$. For the input of this linear layer, we take patch and `[CLS]` output features from four evenly spaced layers of the backbone, not applying the last layer normalization.

**Detection** For detection, we evaluate on COCO (Lin et al., 2014), reporting the average precision (AP) score. We train upon the Plain-DETR implementation (Lin et al., 2023) using the configuration provided in the official repository. More specifically, we train a RPE DETR model during 12 epochs with a linear rate of 0.0002 and a 1000 steps warmup. Compared to the default configuration, and following Siméoni et al. (2025) evaluation framework, we keep the ViT encoder frozen.

When training with DINOv2 and DeiT-III models, we follow the default configurations provided in the official repository, modified to add biases in the attention and to specialize layers or blocks.

When training with CaiT, we follow the default architecture provided in Touvron et al. (2021) and train the models using the self-supervised framework of DINOv2. We use the same hyperparameters as DINOv2 experiments.

## A.4 SPECIALIZATION OF NORMALIZATION LAYERS IN DEIT-III

We report in Fig. 11 the impact of the specialization of the normalization layers when using DeiT-III pre-training strategy. Similar to the case of DINOv2, the average `[CLS]`-patch cosine similarity significantly reduces when employing the specialized normalization, showing the disentanglement effect.

## A.5 LORA APPROXIMATION

As the parameters increase can be a bottleneck for training efficiency, we explore the use of Low-Rank Adaptation (LoRA) (Hu et al., 2022) techniques to reduce the number of trainable parameters while maintaining performance. Additionally, we hypothesise that `[CLS]` and patches representations share common features. Hence we consider `[CLS]` stream as a specialization of patches stream

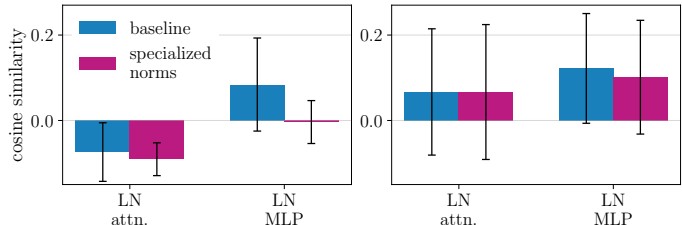

(a) [CLS] -patch cosine similarity    (b) all-all patch cosine similarity

Figure 11: **Specialization of normalization layers.** Mean and standard deviation of the cosine similarity computed between (a) the [CLS] and all patches and (b) all patch to all patches. The average is computed over 1k images and all model blocks. We compare post-normalization statistics between the standard architecture ('Baseline') and the model after normalization specialization ('Specialized norms'). 'LN' stands for LayerNorm and 'LS' for LayerScale.

instead of a complete different stream. Then, for a layer $f$ that we choose to specialize, we compute the operation on the class token $x_{cls}$ as the sum of the patches layer $f_{patch}$ and a low-rank adaptation (LoRA) decomposition $f_{cls}^{(r)}$ of rank $r$.

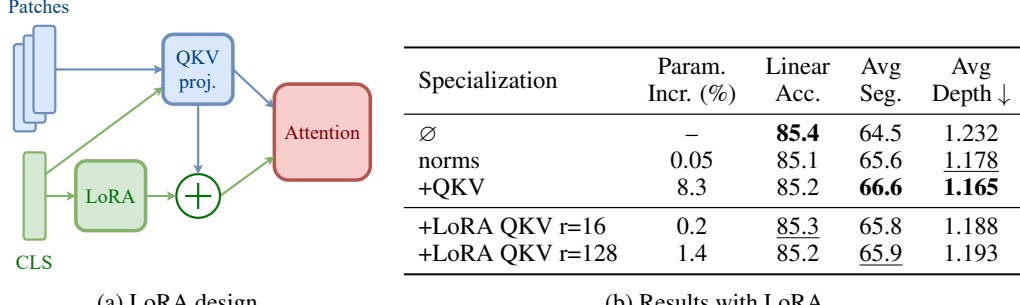

| Specialization | Param. Incr. (%) | Linear Acc. | Avg Seg. | Avg Depth ↓ |
|---|---|---|---|---|
| ∅ | – | **85.4** | 64.5 | 1.232 |
| norms | 0.05 | 85.1 | 65.6 | 1.178 |
| +QKV | 8.3 | 85.2 | **66.6** | **1.165** |
| +LoRA QKV r=16 | 0.2 | 85.3 | 65.8 | 1.188 |
| +LoRA QKV r=128 | 1.4 | 85.2 | 65.9 | 1.193 |

(a) LoRA design        (b) Results with LoRA

Figure 12: **LoRA impact.** (a) Visualization of LoRA design : [CLS] as an approximation of patches. (b) Performance metrics and parameter increase for different LoRA configurations (rank 16 and 128) during first third of the model. In all cases, the normalization specialization described in Sec. 4.3 is applied, corresponding to 'norms' row.

We conduct experiments in which we specialize normalization layers and the QKV projections with LoRA approximations of ranks 16 and 128 (over an embedding dimension of 1024). The results presented in Fig. 12b shows improvements (+0.2 and +0.3 in segmentation tasks) over specializing only the normalization layers, while adding a limited number of parameters (+0.15% and +1.35% respectively). We leave further investigations as future work.

## A.6 NORMALIZATIONS ARE NEEDED

Additionally to the specialization experiments we produced in Sec. 4.5, we also conduct an experiment specializing QKV projection during the first third of the model, *but not the normalization layers*. We plot the results of this experiment in Table 5 compared to the baseline and to our best model specializing normalization layers and QKV projection during third of the model. We observe that specializing only QKV projections brings little improvement over the baseline, e.g. +0.2 mIoU pt in average on segmentation tasks. This shows that specializing the normalization layers is crucial for best performance.

## A.7 ADDITIONAL RESULTS ON DEIT-III

We report in Fig. 13 the performance curves on VOC segmentation task during the training of ViT-B and ViT-L models when following DeiT-III. We observe that the performance reaches its peak in the middle of the pre-training, then drops significantly towards the end. We attribute this behavior

Table 5: **Importance of specializing norms.** Performance of for different layer specialization (Spec.) strategies applied on the first third of the transformer blocks). Normalization layers are specialized in all blocks. Baseline is a ViT-L DINOv2 with attention bias.

| Model | Linear Acc. | Avg. Seg. | Avg. Depth ↓ |
|---|---|---|---|
| Baseline | **85.4** | 64.5 | 1.232 |
| Specialized norms | 85.1 | 65.6 | 1.178 |
| Specialized norms & QKV proj. | 85.2 | **66.6** | **1.165** |
| Specialized QKV proj. | 85.4 | 64.7 | 1.211 |

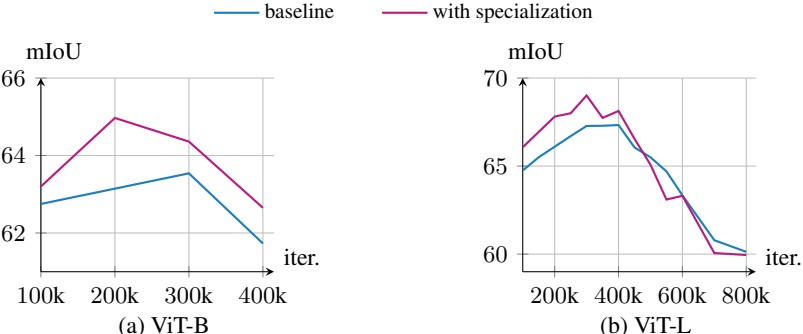

Figure 13: **DeiT-III training evolution.** We visualize VOC segmentation performance (mIoU) throughout training for (a) ViT-B and (b) ViT-L pre-trained with DeiT-III ('baseline') and when adding our layer specialization.

to the lack of a local loss to drive dense performances. We observe a significant gain with our specialization in the first half of the training, but the gains are then diluted in the drop, particularly in the case of ViT-L.

## A.8 OTHER QUALITATIVE RESULTS

We produce in Fig. 14, 15 and 16 more qualitative results when pre-training the model following DINOv2 with the vanilla architecture, four registers or attention bias and when integrating our specialization. Each figure shows the first three components, computed with the patch features, and mapped to RGB. In all cases, we observe that the specialization helps to produce more precise patch features with less artifact. For instance, we invite the reader to pay attention to the back of the dog (first row), where the artifacts visible in the original pre-training are notably reduce with our specialization.

## A.9 WRITING DETAILS

We have used Large Language Models (LLMs) to help write and proofread this paper. More specifically, they have helped to rephrase some parts of the text, propose synonyms, and check the grammar. We have carefully checked all the outputs of the LLMs to ensure that they are accurate and faithful to our work.

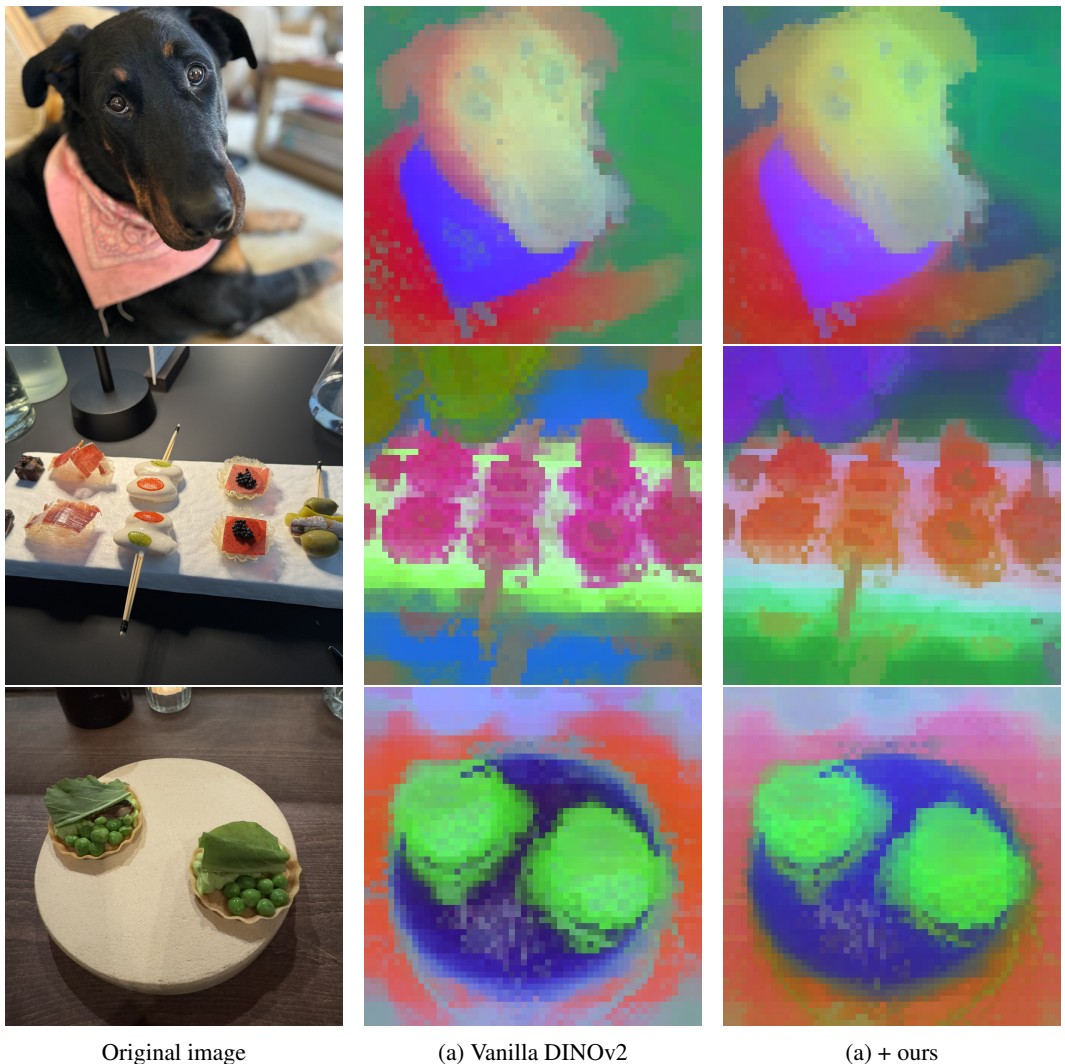

| Original image | (a) Vanilla DINOv2 | (a) + ours |

Figure 14: First PCA components of model outputs in RGB. Specialization of normalizations and QKV projections is made during 1/3 of the model. ViT-L with vanilla DINOv2.

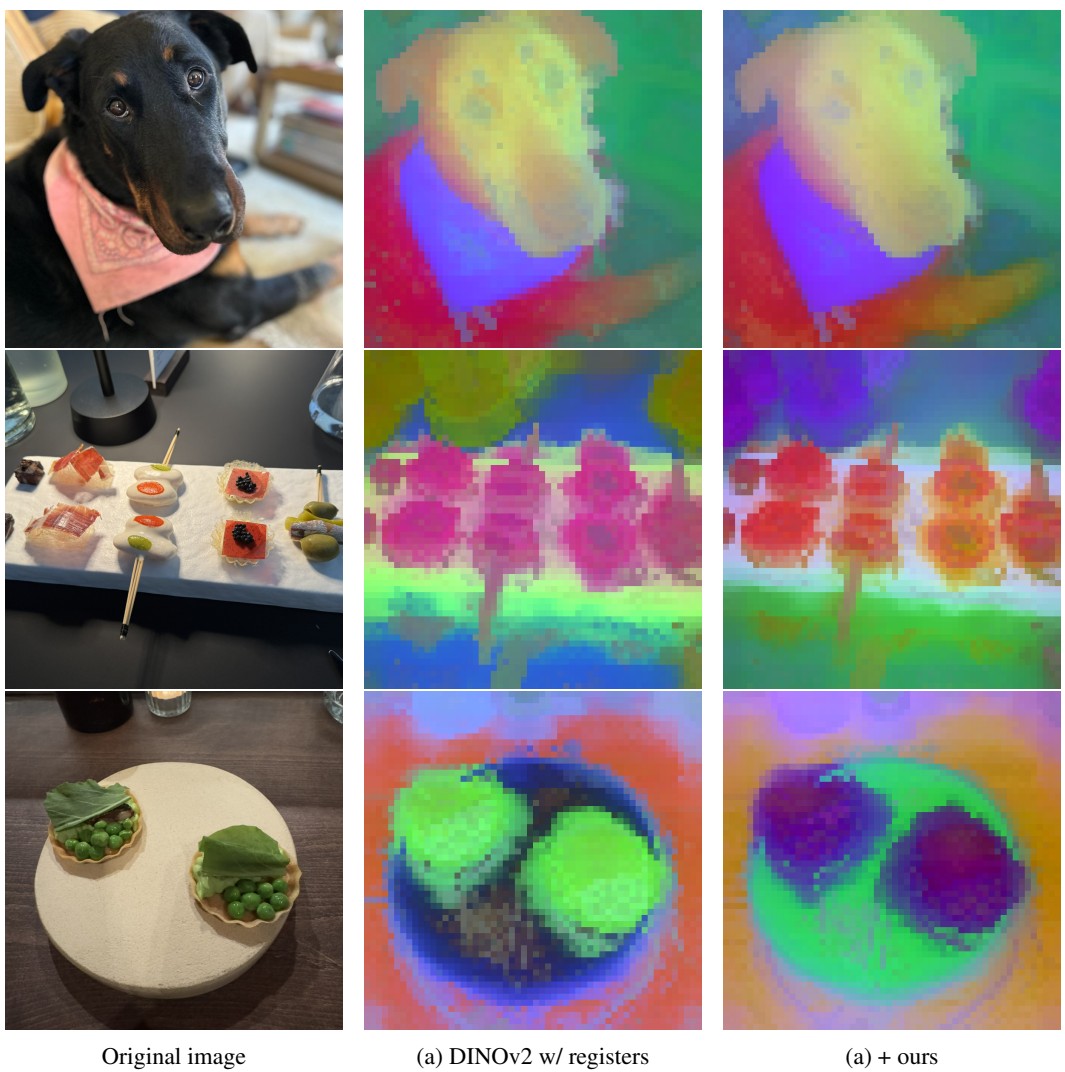

| Original image | (a) DINOv2 w/ registers | (a) + ours |

Figure 15: First PCA components of model outputs in RGB. Specialization of normalizations and QKV projections is made during $1/3$ of the model. ViT-L DINOv2 with four registers.

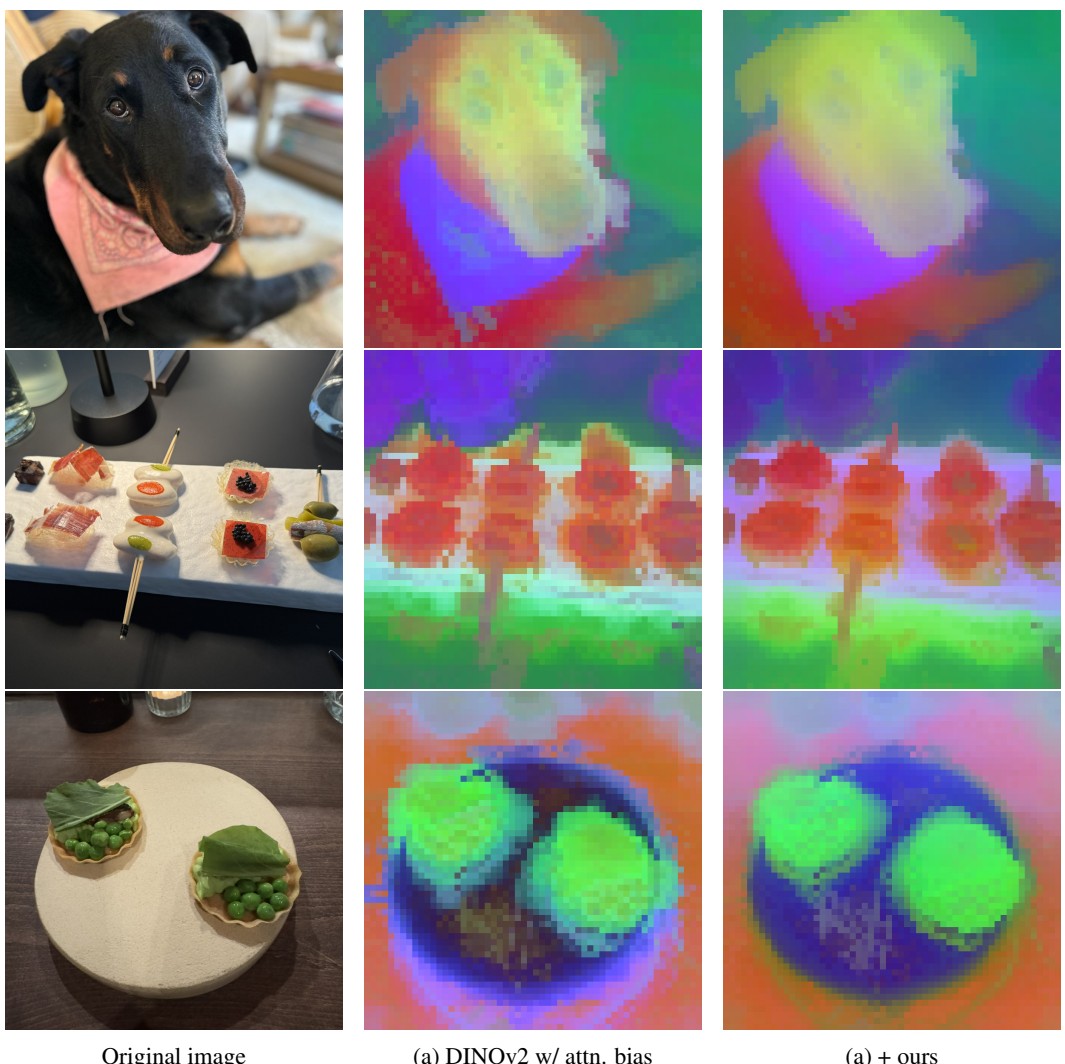

| Original image | (a) DINOv2 w/ attn. bias | (a) + ours |

Figure 16: First PCA components of model outputs in RGB. Specialization of normalizations and QKV projections is made during $1/3$ of the model. ViT-L DINOv2 with attention bias.

