# OpenReview forum: "Revisiting [CLS] and Patch Token Interaction in  Vision Transformers"
_ICLR.cc/2026/Conference — ICLR 2026 Poster_

### Official Review · Reviewer_3jym · 2025-10-17

**Soundness:** 3
**Presentation:** 2
**Contribution:** 3
**Rating:** 6
**Confidence:** 3

**Summary:**

This paper investigates the interaction dynamics between the [CLS] token and patch tokens in Vision Transformers (ViTs), particularly under different pre-training strategies. The authors identify that standard ViT architectures treat both token types identically, despite their distinct roles—[CLS] for global representation and patches for local features. Through empirical analysis, they reveal that normalization layers implicitly differentiate these tokens. Building on this insight, they propose architectural modifications that explicitly specialize the processing paths for [CLS] and patch tokens, especially in normalization and QKV projection layers. Their approach improves dense prediction tasks (e.g., segmentation and depth estimation) by up to 2 mIoU points, with minimal parameter overhead and no increase in computational cost.

**Strengths:**

* The proposed specialization strategy is simple yet impactful, improving dense prediction performance with only ~8% increase in parameters and no additional FLOPs.

* The authors conduct extensive evaluations across model scales (ViT-B, L, H), training paradigms (DINOv2, DeiT-III), and tasks (segmentation, depth estimation, classification).

**Weaknesses:**

* Lack of theoretical justification. The paper is empirically strong but could benefit from deeper theoretical grounding to explain why specialization improves representation quality.

* To highlight the effectiveness of the proposed layer specialization, while PCA visualizations are provided (Figure 1), more quantitative metrics on feature quality (e.g., clustering scores) could strengthen the argument.

**Questions:**

* Why the proposed specialization strategy hurts the performance of image classification (Table 6(c) and Figure 9(a)) but benefits dense prediction tasks? Especially, the [CLS] token, which your method focus on, seems to be more important for image classifiction rather than dense prediction tasks.

---

> ### Author Response · Authors · 2025-11-20
>
> We thank the reviewer for their valuable and insightful comments. We address each of their questions below.
>
> >#### Lack of theoretical justification. The paper is empirically strong but could benefit from deeper theoretical grounding to explain why specialization improves representation quality.
> As discussed in [1], theoretically characterizing what constitutes a “good” learned representation remains a challenging open problem in deep learning, due to the lack of formal metrics for representation quality and the highly empirical nature of modern deep models. In line with this observation, our work is primarily experimental and aims to provide insights grounded in empirical evidence. Within this scope, we have made an effort to systematically support all claims through both qualitative analyses and quantitative evaluations across multiple tasks. Our objective is to help build intuition about why specialization improves representations, even if a complete theoretical ground remains out of reach at present. We believe this combination of empirical rigor and interpretability aligns well with the current understanding of representation learning.
>
> >#### To highlight the effectiveness of the proposed layer specialization, while PCA visualizations are provided (Figure 1), more quantitative metrics on feature quality (e.g., clustering scores) could strengthen the argument.
> In order to demonstrate the quality of the features of our model, we present both qualitative results in the form of PCA visualization and quantitative results on various downstream tasks including image classification, semantic segmentation and depth estimation. Indeed, those evaluations are performed using a single linear layer on top of the frozen features, which allows us to directly assess the quality of the features.
> >####
> Additionally, following the suggestion of reviewer rfYY, we add additional evaluations on COCO detection in Tables 2 and provide results on ViT-L here.
> COCO detection:
> | Model  | without ours | with ours |
> | ------------- |:-------------:|:-------------:|
> | DINOv2 ViT-L vanilla | 45.6   | **46.8**    |
> | DINOv2 ViT-L register  | 46.4    | **46.8**   |
> | DINOv2 ViT-L attn bias    | 46.0        | **48.2**  |
> We observe that for all ViT-L models, integrating our separation consistently boosts detection performance, with gains of up to 2 points in the case of attention bias training. These results confirm that our specialization strategy improves patch-level representations, beneficial for dense prediction tasks.
>
>
> >#### Why the proposed specialization strategy hurts the performance of image classification (Table 6(c) and Figure 9(a)) but benefits dense prediction tasks? Especially, the [CLS] token, which your method focus on, seems to be more important for image classification rather than dense prediction tasks.
> We thank the reviewer for the question. Our method is designed to enhance the overall quality of image representations by stabilizing the optimization of both CLS and patch tokens. Current training protocols, such as DINOv2 and DeiT, are heavily tuned to optimize global representations—primarily via the CLS token—which accounts for their strong performance on classification tasks. However, prior work [2] has shown that during training, DINOv2 gradually compromises patch-level quality: while the global CLS representation continues to improve, patch features deteriorate.
> Our specialization strategy directly addresses this phenomenon. By encouraging patch tokens to retain their distinct roles and by decoupling their optimization from that of the CLS token, we prevent this collapse of patch quality. Importantly, our method has little impact on the CLS token itself, which explains why classification performance remains largely stable. At the same time, preserving strong and diverse patch-level features yields consistent gains on dense prediction tasks, where spatial detail is essential.
>
> [1] Bengio, Yoshua, Aaron Courville, and Pascal Vincent. "Representation learning: A review and new perspectives." IEEE transactions on pattern analysis and machine intelligence 35.8 (2013): 1798-1828.
>
> [2] Siméoni, Oriane, et al. "Dinov3" arXiv preprint arXiv:2508.10104 (2025).

---

> ### Comment · Reviewer_3jym · 2025-11-27
> **Post Rebuttal Response**
>
> Thanks for the clarification. I will keep my positive rating due to the simplicity of the proposed design and the empirically strong performances on dense prediction tasks. However, I cannot raise the rating further, as the work provides limited conceptual insights or broader inspiration beyond the architectural specialization, and thus did not generate strong excitement for me.

---

### Official Review · Reviewer_HUNB · 2025-10-20

**Soundness:** 4
**Presentation:** 3
**Contribution:** 3
**Rating:** 6
**Confidence:** 4

**Summary:**

This paper proposes an architecture specialization method to treat the CLS token and patch tokens distinctively in the ViT models, resulting in a disentangled distribution of two types of tokens. The sufficient and reasonable experiments support the methods and hyper-parameter configurations. The visualizations also coincide with the effectiveness in dense prediction tasks.

**Strengths:**

1. The method section, along with comprehensive experiments, is impressive. The analyses are comprehensive with both qualitative and quantitative results. The statements are validated by the experiments and clearly support the proposed Specialization for the CLS token and patch tokens.

2. The proposed method is compatible with existing methods (register tokens and attention bias), further improving the overall results of ViT across multiple benchmarks, as shown in Table 2. And it can also stabilize downstream tasks training, as shown in Figure 9. The above experiments stand for the practical application of the proposed method in the general vision area.

3. Table 1 demonstrates that the performance gain does not simply come from the increasing parameters, but rather an intricate design through complete analyses.

**Weaknesses:**

1. The motivation is unclear at the beginning, such as in the Abstract, Introduction, and Methods. The motivation starts with rethinking the model architecture design, while the specific target is to improve the downstream performance. I would recommend starting with the tasks and the subsequent modifications that could leverage the disentangled distribution to improve the performance.

2. The analysis and explanation of "Dimension separation" from L206 is not clear enough. For example, how to better understand Figure 4?  Is the other axis the "token index"? More explanation and a clear figure are required.

3. What is the training strategy for ViT-H? This detail is missing.

4. How to explain the visualization impact? For example, Figure 1 demonstrates the two distinctive results by register token and attn bias. However, when applying the proposed specialization method, the visualization becomes similar in general. Can it be explained that the specialization is more fundamental and effective in changing the feature distribution? In addition, from Figures 14-16, the specialization method produces more ambiguous and blurred visual details. Why could these features contribute to better dense prediction tasks?

**Questions:**

The questions have been listed in the weakness part. I would consider further raising the score if the authors could address my concerns.

---

> ### Author Response · Authors · 2025-11-20
>
> We thank the reviewer for their valuable and insightful comments. We address each of their questions below.
>
> >#### The motivation is unclear at the beginning, such as in the Abstract, Introduction, and Methods. The motivation starts with rethinking the model architecture design, while the specific target is to improve the downstream performance. I would recommend starting with the tasks and the subsequent modifications that could leverage the disentangled distribution to improve the performance.
> We thank the reviewer for the suggestion, but we respectfully disagree. The goal of this work is not to optimize performance on a few specific downstream tasks, but rather to address a structural limitation of the Vision Transformer architecture—namely, the joint processing of CLS and patch tokens—which influences the learning dynamics and hinders the emergence of strong image representations.
> To this end, we conduct an in-depth analysis of the interactions between CLS and patch tokens, demonstrating the friction that arises from their joint processing. We then propose weight specialization as a principled way to mitigate this issue. To validate its effectiveness, we present a comprehensive set of empirical results that goes beyond downstream task performance. In Section 4.3, we show that token specialization reduces CLS–patch interference, and in Figure 9, we further demonstrate that it leads to improved learning dynamics, as reflected by the widening performance gap between the baseline and our method during training.
>
>
> >#### The analysis and explanation of "Dimension separation" from L206 is not clear enough. For example, how to better understand Figure 4? Is the other axis the "token index"? More explanation and a clear figure are required.
> We thank the reviewer for pointing it out. Following the reviewer’s suggestion, in the revised manuscript, we improve both the figure caption and the accompanying text to better guide the reader. In particular, we now explicitly clarify which tokens are selected :
> “
> Dimensions with biggest magnitudes early (a), in the middle (b), at the end (c) of the model for [CLS] and $5$ patches with the highest magnitudes in the selected dimensions.
> “
> We also provide an example to interpret the visualization :
> “
> We observe that some specific dimensions are leveraged only by a certain token type. For example, in Fig 4.c, the $2$nd dimension presents large magnitudes for patches and almost none for [CLS]. Moreover, the deeper we are in the model, the fewer token types share dimensions. This enables normalization layers to perform distinctive operations.
> “
> We hope these changes make the concept and the figure substantially easier to understand.
>
> >#### What is the training strategy for ViT-H?
> We thank the reviewer for the remark; we have updated the Table 2 caption to clearly describe the training strategy used for ViT-H, which is in fact DINOv2 framework with attention bias.
>
> >#### How to explain the visualization impact?
> Register tokens and attention bias strategies address the issue of localized artifacts (and specifically high-norm tokens) that arise in the attention mechanism. These techniques help prevent certain patch tokens from becoming outliers but, in our view, exert only a limited influence on the feature distribution. In contrast, the proposed specialization strategy selectively disentangles the distributions of the [CLS] token and patch tokens, thereby better preserving both the quality and diversity of features throughout training.
> Regarding Figures 1 and 14–16, although the specialized features may appear more homogeneous, they seem to contain fewer systematic artifacts (e.g., clearer separation between sea and sky, more consistent dog fur textures, or more coherent plate boundaries in food images). We hypothesize that this reduction of high-frequency artifacts leads to features that generalize more effectively in dense prediction tasks.

---

> > ### Comment · Reviewer_HUNB · 2025-11-25
> > **Post Rebuttal Response by Reviewer HUNB**
> >
> > Thanks for the response. The author addressed my concerns. The motivation and presentation are clear in the revised version. The additional experiments also support the effectiveness.
> >
> > I have read through all the other reviews and responses by the author. I have raised the score. Good luck!

---

### Official Review · Reviewer_rfYY · 2025-10-25

**Soundness:** 4
**Presentation:** 4
**Contribution:** 3
**Rating:** 6
**Confidence:** 4

**Summary:**

The paper analyzes how Vision Transformers (ViTs) process the global class token [CLS] and local patch tokens and argues that treating them identically causes friction for dense prediction tasks. Building on observations about token dynamics and normalization, the authors propose layer specialization: in selected blocks they use separate parameters for [CLS] vs. patch tokens (e.g., distinct LayerNorms and Q/K/V projections) while keeping attention shared, so information can still flow. On ImageNet pretraining (supervised DeiT-III and self-supervised DINOv2), they report essentially unchanged ImageNet classification accuracy but around +2 mIoU improvements on semantic segmentation and better depth RMSE, with only a small (~8%) parameter increase. The approach relates to earlier ViT foundations (ViT, DeiT-III, DINO/DINOv2) and to fixes for high-norm/outlier tokens (e.g., Registers) while differing from CaiT (which adds dedicated class-attention layers) by specializing within standard blocks rather than inserting new class-attention stages.

Summary of the review: The paper presents a clear and well-motivated analysis of why identical processing of global and local tokens can limit dense prediction, proposing a simple yet effective solution via token-type-specific normalization and projections. The method is elegant, lightweight, and yields consistent improvements (~+2 mIoU) across models and pretraining setups. However, the novelty is somewhat incremental given overlap with CaiT’s class-attention, and broader validation (e.g., detection, in-context tasks) plus efficiency reporting would strengthen the case. Overall, it’s a technically solid and well-executed contribution with moderate originality, justifying a rating of 6 for clear insight and strong empirical support.

**Strengths:**

1) Clear problem framing: Why identical processing of global/local tokens can hinder dense tasks is well-motivated and consistent with literature on token dynamics and normalization.

2) Simple, effective change: Token-type-specific LayerNorm/QKV yields consistent gains without changing attention/FLOPs; minimal engineering overhead.

3) Robust evaluation: Results across pretraining regimes (DINOv2, DeiT-III) and model sizes; improvement magnitude (~+2 mIoU) is meaningful for segmentation.

**Weaknesses:**

1. Novelty overlap: Since CaiT[1] already introduces class-attention (explicitly specializing how [CLS] aggregates information), authors should include a head-to-head comparison or discussion clarifying differences/advantages (e.g., same backbone size/recipe; dense-task transfer).

2. Limited Task coverage: Add a few more tasks, like the hummingbird evaluation for in-context vision understanding, object detection/instance segmentation (e.g., COCO) to show benefits generalize beyond conventional segmentation/depth.

3. Training efficiency: Report training memory/time impact of the ~8% parameter increase (batch size, wall-clock), not just inference parity.

[1] Touvron, H., Cord, M., Sablayrolles, A., Synnaeve, G., & Jégou, H. (2021). Going deeper with Image Transformers. arXiv [Cs.CV].  http://arxiv.org/abs/2103.17239

**Questions:**

1. CaiT vs. this work: On equal pretraining and decoder setups, how does your method compare to CaiT backbones for dense tasks? Any synergy if both are combined?

2. Registers vs. specialization: With identical training, which helps dense tasks more on its own—Registers or your specialization? Any redundancy when combined?

3. Detection results: Do COCO detection/instance-seg metrics show similar benefits?

4. In-Context Visual Understanding: Does the model gain any benefits on the recent proposed Hummingbird evaluation [1] ( implemented openly by [2])?

5. Stability & hyperparams: Does specialization require different learning-rate/weight-decay schedules to avoid the minor classification drops you mention in fully-specialized variants?

[1] Balažević, I., Steiner, D., Parthasarathy, N., Arandjelović, R., & Hénaff, O. J. (2023). Towards In-context Scene Understanding. arXiv [Cs.CV]. http://arxiv.org/abs/2306.01667
[2] https://github.com/vpariza/open-hummingbird-eval

---

> ### Author Response · Authors · 2025-11-20
>
> We thank the reviewer for their valuable and insightful comments. We address each of their questions below.
> >#### CaiT vs. this work: On equal pretraining and decoder setups, how does your method compare to CaiT backbones for dense tasks? Any synergy if both are combined?
> We thank the reviewer for the suggestion and provide here a comparison between our approach and CaiT (class attention). We agree that both methods share the high-level idea of treating CLS and patch tokens differently. However, the mechanisms differ in an important way: CaiT introduces two sequential processing stages—first encoding patch tokens, then aggregating them into the CLS—whereas our specialization method creates two parallel pathways throughout the network. Our design allows the CLS to gather information from patch tokens at every layer, leveraging the progressively richer levels of granularity learned across the hierarchy.
> Following the reviewer’s recommendation, we conduct experiments with CaiT using DINOv2 ViT-L, both with and without attention bias. Please note that we consider a setting that is different from the initial setup of CaiT, which was originally introduced and evaluated  in a supervised setup. Following CaiT, we remove the CLS from the main transformer blocks and add two class-attention layers at the end to aggregate the final patch features. The results, shown in Table 3 of the revised paper (and added here), indicate that our specialization method consistently outperforms CaiT across all benchmarks, regardless of the presence of attention bias.
>  | Method              | Classif. (ImNet) | Seg. (ADE) | Seg. (City) | Seg. (VOC) | Depth ↓ (KITTI) | Depth ↓ (NYU) | Depth ↓ (SUN) | Detection (COCO) |
> |---------------------|------------------|------------|-------------|------------|------------------|----------------|----------------|-------------------|
> | ∅                   | **85.3**         | 45.7       | 64.2        | 82.1       | 2.868            | 0.389          | 0.410          | 45.6              |
> | CaiT                | 84.0             | 43.5       | 63.7        | 80.8       | 2.968            | 0.397          | 0.421          | 43.1              |
> | Ours                | **85.3**         | **47.3**   | **66.6**    | **83.7**   | **2.787**        | **0.369**      | **0.390**      | **46.8**          |
> >####
> | Method              | Classif. (ImNet) | Seg. (ADE) | Seg. (City) | Seg. (VOC) | Depth ↓ (KITTI) | Depth ↓ (NYU) | Depth ↓ (SUN) | Detection (COCO) |
> |---------------------|------------------|------------|-------------|------------|------------------|----------------|----------------|-------------------|
> | Attn. bias    | **85.4**         | 46.2       | 65.2        | 82.2       | 2.917            | 0.373          | 0.406          | 46.0              |
> | Attn. bias + CaiT   | 85.2             | 45.3       | 65.7        | 82.0       | 2.882            | 0.374          | 0.400          | 46.8              |
> | Attn. bias + ours   | 85.2             | **48.4**   | **67.4**    | **84.0**   | **2.739**        | **0.362**      | **0.393**      | **48.2**          |
> >#### Because CaiT and our specialization method differ fundamentally—CaiT processes CLS and patch tokens sequentially, whereas our approach processes them in parallel—it is not straightforward to determine a meaningful way to combine the two. We would be happy to explore such a combination, but we would appreciate more specific guidance from the reviewer on the intended integration.

---

> ### Author Response · Authors · 2025-11-20
>
> >#### Registers vs. specialization: With identical training, which helps dense tasks more on its own—Registers or your specialization? Any redundancy when combined?
> We thank the reviewer for the question. In Table 2 (rows 2–4) of the submission, we report the benchmark performance when applying our specialization method, registers, or both on a vanilla DINOv2 ViT-L. The results show that our specialization achieves comparable ImageNet classification performance to using registers alone, while providing substantially better results on all other benchmarks—yielding an average improvement of 1.6 mIoU on semantic segmentation and a relative gain of 3.2% on depth estimation. However, combining specialization with registers does not provide additional benefits. Although the combination outperforms using registers alone, best results on Cityscapes semantic segmentation and KITTI depth estimation benchmarks are obtained with specialization alone.
>
> >#### Limited Task coverage: Add a few more tasks, like the hummingbird evaluation for in-context vision understanding, object detection/instance segmentation (e.g., COCO) to show benefits generalize beyond conventional segmentation/depth.
> We thank the reviewer for the suggestion. As recommended, we evaluate our models on the widely used COCO detection benchmark. Specifically, we train the out-of-the-box Plain-DETR [1] detector without hyperparameter tuning and keeping the features backbone frozen. We integrate results with all models in Table 2 and report below performance for our ViT-L models trained with the vanilla strategy, with four registers, or with attention bias, both with and without our specialization.
> >####
> | Model  | without ours | with ours |
> | ------------- |:-------------:|:-------------:|
> | DINOv2 ViT-L vanilla | 45.6   | **46.8**    |
> | DINOv2 ViT-L register  | 46.4    | **46.8**   |
> | DINOv2 ViT-L attn bias    | 46.0        | **48.2**  |
> | DINOv2 ViT-B attn bias    | 39.6       | **39.8**  |
> | DINOv2 ViT-H attn bias    | **49.9**        | 49.5  |
> | DEIT ViT-B attn bias    | **32.8**        | 32.5 |
> >####
> We observe that for all ViT-L models, integrating our separation consistently boosts detection performance, with gains of up to 2 points in the case of attention bias training. For other architectures, the impact is smaller; we believe that careful hyperparameter tuning could further increase the benefit. These results confirm that our specialization strategy improves patch-level representations, beneficial for dense prediction tasks.
> We hope that the complementary results we have provided address the reviewer’s concerns. While we faced challenges running the Hummingbird evaluation within the scope of this rebuttal, we will investigate these evaluations further if the reviewer deems them critical for strengthening the empirical evidence.
>
> >#### Stability & hyperparams: Does specialization require different learning-rate/weight-decay schedules to avoid the minor classification drops you mention in fully-specialized variants?
> Throughout our experiments, we always use the hyperparameters provided by the official repositories for each framework. It turns out that our method works out-of-the-box and does not require different learning-rate or weight-decay schedules. The training also remains stable. We believe that further tuning could help to avoid the minor classification drops but opted out in order to ensure fair comparisons to baselines and isolate the impact of our token specialization method.

---

### Author Response · Authors · 2025-11-20

We thank all reviewers for their time and thoughtful feedback, and we appreciate the positive assessments of our work.

All reviewers have noted that the paper is technically solid, clearly written, and supported by strong empirical evidence. We are glad that our analysis of token interactions in Vision Transformers was found to provide a clear and well-motivated explanation for why identical processing of global and local tokens can hinder the model from learning good representation.

We also appreciate the recognition that our proposed specialization mechanism is simple and lightweight, requiring minimal architectural changes while consistently improving performance across model sizes (B/L/H) and pretraining frameworks (DINOv2, DeiT-III). Its ability to yield meaningful gains in segmentation and depth estimation without additional computational cost—and to integrate seamlessly with complementary approaches such as register tokens and attention bias—was likewise highlighted by reviewers.

We address all reviewer comments in detail in the individual responses below and upload a new version of the paper, with highlighted changes.

---

### Meta-Review · Area_Chair_iTVn · 2026-01-05

**Summary:**

This paper shares the empirical finding that a natural specialization between the [CLS] and image tokens in ViT architectures exists and is beneficial to downstream dense prediction tasks. Based on it, it proposes a simple architectural change of specializing parts of the ViT networks for each of these types of tokens with differing network weights. This, simple change when combined with other token bias mechanisms such as registers and attention biasing, results in modest, but consistent increases in the accuracy of downstream dense prediction tasks, which maintaining the accuracy of global prediction tasks.

Three reviewers provided a final ratings of 6, 6, and 6. All reviewers appreciated the paper's novel empirical insight, practical and simple method of separating the two types of tokens, the presentation of paper and its comprehensive experiments. However they showed marginal excitement in the work's impact, because of "limited conceptual insights or broader inspiration beyond the architectural specialization".

All things considered, the AC feels that the contribution of the work slightly surpasses its shortcomings and hence is leaning towards acceptance, but it would be OK if this work were rejected.

**Reviewer Concerns:**

Concerns addressed:
* Comparisons to CaiT[1]
* Evaluation on more downstream tasks
* Training stability
* Improved presentation

Concerns not addressed:
* Lack of theoretical justification of the proposed approach and the observed empirical insight

**Reviewer Scores:**

1. Reviewer rfYY (Rating: 6: marginally above the acceptance threshold. But would not mind if paper is rejected)

The reviewer's primary concerns were (a) comparisons to CaiT[1] and limited novelty, (b) evaluation on a few more tasks, e.g., object detection/instance segmentation and (c) reporting training efficiency and stability. All the reviewer's concerns were mostly addressed. The reviewer is likely to have maintained their original rating.

2. Reviewer HUNB (Rating: 6: marginally above the acceptance threshold. But would not mind if paper is rejected)

The reviewer's main concerns were around improving the presentation of the paper and adding additional clarification. The authors' rebuttal addressed their concerns and they raised their original score to a final value of 6.

3. Reviewer 3jym (Rating: 6: marginally above the acceptance threshold. But would not mind if paper is rejected)

The reviewer's primary concerns were (a) the lack of theoretical justification of the proposed method and (b) more empirical evidence to highlight the effectiveness of the proposed layer specialization. The authors rebuttal partially addressed the reviewer's concerns and they provided a final rating of 6.

---

### Decision · Program_Chairs · 2026-01-26

Accept (Poster)